# Communicating Sustainable Responsible Investments as Financial Advisors: Engaging Private Investors with Strategic Communication

Nadine Strauß 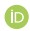

Oxford Sustainable Finance Programme, Smith School of Enterprise and the Environment, University of Oxford, Oxford OX1 3QY, UK; nadine.strauss@smithschool.ox.ac.uk

**Abstract:** Although sustainable responsible investing (SRI) has increasingly become popular on the financial markets, the potential of raising capital from private investors for sustainable development has not been efficiently seized thus far. The lack of knowledge and training about SRI by financial advisors has often been identified as one of the main reasons for this investment gap. In order to accelerate the role of financial advisors as change agents for SRI, this study proposes several strategic communication interventions that advisors could employ in their advisory talks to raise more attention and engagement among private investors for SRI. The interventions proposed are oriented on the 5A model of SRI decision making by Herwig Pilaj and drawn from an interdisciplinary literature review on sustainability, communication, and attitudinal and behavioral change. The results provide a perspective and practical guide for financial advisors on how to effectively communicate SRI to private investors. Limitations and areas future research are discussed.

**Keywords:** sustainable responsible investments; financial advisors; strategic communication; message framing; decision-making; literature review

## 1. Introduction

Sustainable responsible investing (SRI) has largely entered mainstream on the financial markets [1,2]. Since the Paris climate summit 2015, the number of assets under management that follow environmental, social, and governance (ESG) criteria in Europe has reached more €882 billion by the third quarter in 2020, accounting for 9.3% of total European assets, according to Morningstar [3]. Furthermore, while Europe makes up 82% of the sustainable global fund universe [4], it is expected that up to 57% of mutual funds in Europe will adhere to ESG criteria by 2025 [5]. It appears that the majority of banks and financial institutions, particularly in Europe, have become aware that climate change does not only pose an environmental risk but will increasingly pose a financial risk in terms of stranded assets [6], unaffordable insurance premiums due to climate risks [7], or climate policy regulations (e.g., carbon tax) [8]. In light of these financial climate risks, more than 3000 financial institutions have signed the Principles of Responsible Investors by the end of 2020, pledging to integrate ESG criteria into investment decisions and practice [9]. Accordingly, sustainable fund assets have amounted to $1.26 trillion worldwide by the third quarter of 2020, offering more than 3700 sustainable funds globally [4].

However, although sustainable investments have become more popular on the financial markets, there is still a large investment gap to be filled to reach the $4.5 trillion additional investments needed per year to reach the United Nations Sustainable Development Goals (SDGs) by 2030 [10]. Scholars have argued that there is a lack of involvement of private investors with sustainable investments and that there is large potential in moving capital from individual investors to realize the SDGs [11–13]. Particular high net worth individuals, those who have a net worth exceeding $50 million, seem to be interested in sustainable investments [12]. Similarly, millennials who are expected to inherit more

than US$68 trillion by 2030 in the U.S alone [14] belong to the demographic group (young, liberal, educated, wealthy) that is mostly associated with sustainable investments [15,16]. Yet, while the general awareness of sustainable investments has increased in the past years, the majority of private investors remain uninformed about SRI and hesitate to invest. For example, a global survey among more than 5300 high net worth investors by UBS showed that, while 65% see a need to help create a better planet, only 39% have sustainable investments in their portfolio [17]. Hence, there is a clear attitude–behavior gap [11,18] among private investors regarding SRI.

One factor that might be largely responsible for this gap is unsuccessful or inefficient financial advisory talks between investors and financial advisors regarding SRI [11,19,20]. Yet, most of the financial decisions, such as investing in sustainable funds, are made as a result of advisory talks between financial advisors and private investors. For example, nine out of ten investors surveyed by the UBS say that an advisor impacted their decision to invest sustainably [17]. At the same time, financial advisors have been found to lack knowledge on sustainable investments [11], to have low ethical or sustainable concerns [20], or to even refrain from offering the possibilities of sustainable investments to their clients at all [19,21]. The latter point might soon be less common, given that the European Union (EU) is said to implement the requirement that financial advisors must inform their clients about sustainability risks of their investments [22,23]. Furthermore, given that many banks and financial institutions have expanded their portfolios of SRI (e.g., ETFs, sustainable funds, green bonds), financial advisors might also soon be required to become experts in selling those products to their clients, similar to conventional financial products (see [24]).

Facing this discrepancy between the need to inform private investors about SRI on the one hand and the lack of financial advisory expertise on the other hand, the question arises of how financial advisors could improve their communication about SRI when talking to private investors in order to shift more capital into SDGs. Previous research has argued that the SRI industry needs targeted marketing measures, nudging [25], and effective communication strategies [11,19,26]. More recently, Paetzold and colleagues [20] have identified investment advisors as frame-makers to be a "promising avenue for future research" (p. 215). Similarly, Linciano and colleagues [21] have pleaded for better communication by financial advisors vis-à-vis their clients to counteract misunderstandings and fill information gaps and misperceptions with regard to SRI. Therefore, the financial industry is in need of clear and straightforward strategic communication guidelines that can be employed in financial advisory talks to successfully achieve SRI decisions by private investors.

In the following, I first review the role of financial advisors for sustainable finance, followed by a summary of challenges regarding SRI in general and for financial advisors and private investors in particular. Then, based on a literature review and the 5A model of SRI decision making by Pilaj [25], I theoretically outline how strategic communication interventions can be used to overcome these barriers in order to increase awareness, attention, and engagement of private investors with SRI. I conclude with a reflection on limitations and an outline for future research.

## 2. The Role of Financial Advisors for Sustainable Responsible Investments

Financial advisors have been ascribed a crucial role for capital markets, particularly in the realm of SRI (e.g., [11,20,21,25,27]). Following the definition by Eurosif, the European association for the promotion of sustainable and responsible investments, SRI can be understood as "a long-term oriented investment approach which integrates ESG factors in the research, analysis and selection process of securities within an investment portfolio. It combines fundamental analysis and engagement with an evaluation of ESG factors in order to better capture long term returns for investors, and to benefit society by influencing the behaviour of companies" [28]. Note that there is no overall agreement on the definition of SRI (cf. [20]). In fact, similar terms such as "sustainable investments", "green finance", "sustainable finance", or "ethical and responsible investments" are used across the global

markets for the overall purpose to promote investments that are in line with the SDGs but with various degrees and focus.

Given the ambiguous use of terms and concepts related to SRI and the various investment products that are associated with it, a clear communication about selection criteria, characteristics, and goals of SRI to private investors is strongly needed. Due to their role as diffusion agents, change agents [11,19], or multipliers [21], financial advisors are considered to be gatekeepers (cf. [27]) in informing investors about SRI. Scholars have repeatedly argued that financial advisors could strongly contribute to an increase of sustainable investments, especially among private investors [11,19,20]. Even the High-Level Expert Group on Sustainable Finance by the European Commission (HLEG) has thoroughly discussed the role of financial advisors to accelerate sustainable finance and SRI across Europe [29].

However, although SRI has recently gained popularity among financial market actors, only about 25% of sustainable and responsible investments are held by retail investors [30]. Private investors are said to have difficulties in understanding financial products and lack the experience in making investment decisions (cf. low levels of financial literacy globally [31]). This is even more true when it comes to sustainable investments (e.g., [17,21,32]). Therefore, financial advisors take on a crucial role in not only providing expert information on financial investments but also in giving private investors personal advice regarding SRI. In fact, a recent study has shown that financial advisors are effective in raising financial awareness and literacy among investors [33], and a recent survey by Linciano and colleagues [21] has shown that the majority of private investors (78%) in Italy rely on the advice by financial advisors when seeking information on SRI (for global numbers, see [17]).

What is more, the relationship between financial advisors and private investors can be considered unique, given that usually both have a long term personal relationship [11,33]. Such personal relationships enable advisors to draw a profile of their clients based on the client's financial knowledge, attitudes, behavior, risk aversion, and other relevant financial characteristics which help advisors in suggesting their clients a suitable investment product [11]. What distinguishes financial advisors from other financial actors is that they also represent the personal link between a bank and a client and thus constitute the key point of intervention to increase the demand of sustainable and responsible investment products on the financial markets [11,19]. As a consequence, it has been argued that, if financial advisors would be better informed about SRI and better trained in selling sustainable financial products, a critical amount of private capital could be redistributed into SRI, thereby accelerating the transition towards a more sustainable future [11,19,20,25].

Despite these potentials regarding SRI, it should be noted that financial advisors are usually subject to professional rules and standards in the respective market (e.g., CFA chart holders). For example, they must provide investment recommendations that are customer oriented and in line with the needs of a customer [11]. While, in some cases, this might exclude sustainable investments ex ante, recent developments in financial regulations in Europe might soon require financial advisors to make their clients aware of any sustainability risks associated with financial products [22,23]. Given this new additional regulatory framework about climate risks and the increasing demand for SRI more generally, many banks and financial institutions seek education and training for their employees in the field of sustainable finance [24]. While most of these trainings and courses usually focus on the technical side of sustainable investments (e.g., investment criteria, performance, impact measurement), learning more about the principles of strategic communication that can be employed when talking with clients about SRI could be equally powerful to overcome the challenges that both financial advisors and private investors are still facing with regard to SRI.

## 3. Challenges of Sustainable Responsible Investments

The field of SRI has often been accused of lacking clear definitions, regulations, and measurements [34–36]. This has resulted in widespread criticism in the past, accusing SRI of non-transparency [19], unreliability [24], and complexity [20]. A common critique

is that ESG and sustainability are understood very differently across the financial sector, thus impeding the comparability of sustainable and responsible investment products [34]. For example, the definition of SRI itself can vary considerably across institutions, such as that some might classify SRI based on exclusion criteria (e.g., no armaments, tobacco, or alcohol), while others might select their investments based on their compliance with ESG criteria (with rather ambiguous and sometimes contested scoring methods [37]), and another area of SRI might solely focus on impact investing [38]. Although the latter is often considered as an evolution of SRI [39], this area also suffers from a lack of transparency (e.g., showing impact), a high-risk profile usually associated with impact investments, and the issue of a supply–demand mismatch [40].

Another drawback is that the actual impact of SRI is not being made transparent or can only be made tangible in the long run (e.g., impact on infrastructure, energy, supply chain). Furthermore, in the past, SRI has suffered from the assumption that such investments do not yield satisfying returns compared to conventional investments. Although previous research has repeatedly proven that SRI provides equally, if not even better, returns in some cases (e.g., [41]), some prejudices and misconceptions about SRI still remain. This is partly due to the fact that the performance of sustainable and responsible investment products strongly depends on the above-mentioned selection criteria (e.g., exclusion vs. inclusion; ESG screening) and can thus strongly vary regarding the region [42], the group of investments (e.g., sustainable funds, [43]), or the measurement of single ESG companies (e.g., [44]). However, the Sustainable Finance Taxonomy, as agreed upon by the EU HLEG [29], and more widely used industry standards such as the Task Force on Climate-related Financial Disclosures (TCFD) or the Sustainable Accounting Standards Board (SASB) that promote integrated and transparent climate-related financial information and sustainability reporting, are a good start to overcome some points of the before-mentioned criticism and to set clear guidelines and frameworks regarding SRI for the financial industry and financial advisors in particular. Hence, in order to better inform private investors and market participants about SRI and some of the remaining issues related with the field, financial advisors need to take on their role as diffusion and change agents [11,19], thereby expanding the potentials of SRI on financial markets.

## 4. Challenges of Financial Advisors

Previous research has shown that the communication of financial advisors about SRI with investors is strongly dependent on advisors' expectations regarding the financial returns of SRI, their trust in the providers of SRI products, and the perceived real world impacts of SRI [20]. In turn, if SRI is understood as a marketing stunt, financial advisors are less likely to communicate with their clients about SRI, according to Paetzold and colleagues [20]. In fact, past research has criticized the role of financial advisors as "diffusion agents for ethical investments" ([19], p. 201) and found that they either ignored the existence of SRI or only provided limited adequate information on sustainable investment products (see also [11,12]). While these results might be outdated by now, the lack of financial advisory on SRI might not only be a result of personal preferences of the respective advisor, it could also be a result of an institution's missed opportunity to provide incentives, information, and training related to SRI for advisors [11,19].

Heinemann and colleagues [11] found, based on an interview study, that German financial advisors attested themselves limited knowledge about sustainability, a lack of qualification in the area of SRI, and a missing orientation on sustainable and responsible investment products in their advisory talks with clients. Likewise, the financial advisors stated that there was a lack of demand for SRI on the side of customers. This vicious cycle has been pointed out by other researchers as well (cf. [11,12,19]). On the one hand, private investors do not demand information from their advisors with regard to SRI due to limited awareness and knowledge; on the other hand, advisors do not provide information on SRI because their clients do not request it. While the demand on the side of customers is slowly

picking up more recently [4], the most efficient way to break through this doom loop is for financial advisors to raise more awareness for SRI during their advisory talks with clients.

However, another issue that arises on the side of financial advisors is a "chasm between what matters for investors and their financial advisors" ([20], p. 216). Rather than being influenced by self-transcendent values or ethical concerns, which are important values for pro-environmental behavior (cf. [45]), advisors are more likely to talk about SRI with their clients when the return profile of SRI is high and when they perceive their customers do not have a clear understanding of SRI [20]. Given that the return profile of SRI is comparable to conventional investments by now [41], financial advisors could effectively use their information advantage about SRI vis-à-vis their clients if they received better communication training as well as transparent and clear information about SRI that they could share with their clients [19,27]. What is more, financial advisors usually receive instructions about which investments to sell to their clients by their employers on a regular basis, which in turn determines their renumeration or bonus. Given the increased demand for SRI and the expansion of sustainable investments products on the market, it is likely that these products will make it quickly to the top of such product and incentive lists.

## 5. Challenges of Private Investors

While financial advisors can be trained in the field of SRI and communication, the obstacles to invest in sustainable and responsible investments on the side of private investors also need to be considered. Paetzold and Busch [12] have presented a list of potential barriers that inhibit private investors from SRI, ranging from fear of low financial returns, insufficient corporate ESG reporting, and incoherent SRI data to a mismatch between the investment approach (e.g., exclusion vs. inclusion) and the client's personal values, and a mistrust towards the market, advisors, and SRI products. Keeping those factors in mind, it is worthwhile to note that there are different types of investors that might be more or less inclined to SRI and thus more or less likely to overcome these obstacles more generally. In the realm of SRI, Beal and colleagues [46] distinguish between the rational investor, the consumer investor, and the investment investor. The rational investor is mostly interested in performance and only invests in sustainable investment products if they have a similar return and risk profile as conventional investments. Consumption investors decide to invest in sustainable and responsible investments because it resonates with their behavior. Besides financial performance, the investment decisions of this group are also influenced by external factors such as fashion, peer pressure, or social imitation. The last group is defined as investment investors. They use SRI as a means to change the behavior of companies so that they align with ESG objectives as a means to bring about social change.

Although there are certainly mixed versions of these investor types, one of the major issues to advance SRI among private investors is that the predominant type of investor on the market is the rational investor who mainly seeks abnormal returns (cf. [47]). Furthermore, research has shown that investors who perceive SRI as volatile were less likely to invest in SRI, particularly when their investment horizon was short and if they had experienced substantial financial losses previously [11,12]. What is more, past research implies that most investors of SRI have already held conventional funds before and are seeking SRI to spread risks (e.g., [47]). In general, customer demand for SRI has been reported to be insignificant in the past [11], but an uptick in interest for SRI has been reported since 2019 [4] and has particularly expanded during the Covid-19 pandemic [48].

Another issue on the side of private investors is their lack of knowledge about SRI and their need for financial advice. A large share of investors might have heard about SRI, but they might not be sufficiently knowledgeable about SRI to make investment decisions. This lack of information has been described as a "classic case of asymmetric information" ([27] p. 46) between clients and sellers of sustainable and responsible investment products. Heinemann and colleagues [11] similarly asserted that "private investors are not fully aware of their alternatives and lack orientation in the selection process" (p. 13). Investors might be faced with too much, too difficult, and too confusing information about SRI [12,19,25].

Investment decisions are indeed complex endeavors and are subject to cognitive biases just like any other individual decision-making process (cf. [25]). The drawback is that both information overload and complexity might lead private investors to procrastinate, become indecisive, and defer their sustainable and responsible investment decisions [25].

Even if private investors showed interest in SRI and were determined to invest, a common phenomenon observed with regard to environmental behavior is the attitude–behavior gap [11]. Many private investors, when surveyed, indicate that they are interested in sustainability, but they rarely transform these attitudes into actual investment behavior. Linciano and colleagues [21], for example, reported that, while 40% of investors in Italy indicate that they take environmental and social factors into account when making investment decisions, only 19% of them actually hold sustainable responsible investments. Similar findings stem from a representative survey in Germany [32]. Here, 45% indicate that they would choose sustainable investments as a new investment, but only 19% indicated that sustainability criteria played a role when selecting investments in the past.

## 6. Theoretical Models for Moral and Ethical Investment Behavior

A range of theories and models in the field of environmental sciences has been tested for pro-environmental or ethical behavior, ranging from cognitive dissonance theory (e.g., [49]) to reasoned action theory (e.g., [50]), social cognitive theory (e.g., [51]), or the elaboration l;ikelihood model (e.g., [52]). Similarly, the field of SRI decision-making has investigated how theories of moral behavior can explain moral and ethical investment behavior (e.g., [53–55]). For example, Hofmann and colleagues [54] showed that the multiple attribute utility theory (MAUT) best explains bidding behavior for personnel-promoting companies compared to the theory of planned behavior [56] or the issue-contingent model of ethical decision making in organizations [57]. The authors conclude that ethical investment behavior is not only influenced by the utility of morality but also by the intention to invest and the moral intensity of the investment rather than profitability. However, Nilsson [55] conducted a survey to test a model of expected influential variables on SRI behavior and could show that both pro-social variables and non-altruistic motives (expected financial return) explain SRI.

While such studies try to explain SRI behavior by measuring various individual variables, the shortcoming of these models is that they imply that investors are already aware of SRI. What is more, they ignore the crucial role of financial advisors in SRI decision-making processes and the various steps that lead to financial decision-making over time [11,19,20]. Pilaj [25] has offered a more realistic model of the SRI decision process. The 5A model encompasses five mental steps that take place when engaging with SRI, including (1) activating the need to think about investments, (2) raising awareness for ethical options, (3) forming an attitude towards SRI, (4) making a decision for SRI, and (5) adjusting and monitoring sustainable and responsible investments in the long run. According to Pilaj, each step triggers the risk of barriers, which are similar to the challenges for private investors regarding SRI described above. Pilaj therefore argues for an SRI-friendly choice architecture, such as setting sustainable and responsible investments as default or asking private investors directly whether they are concerned about the ESG performance of their investments. While it will likely become mandatory for financial advisors to make their clients aware of sustainability risks of investments in Europe [22,23], it is less likely that sustainable and responsible investments will become a standard investment portfolio any time soon. Hence, it is still up to the financial advisor to intervene in those five mental steps by means of effective communication strategies to successfully engage private investors with SRI.

## 7. Strategic Communication Interventions in Financial Advisory Talks

The following strategic communication interventions are based on a literature review of studies in the field of psychology, sociology, marketing, economics, and communication that deal with the promotion of sustainable attitudes and behavior. In the first step,

relevant studies that dealt with the topic of SRI, sustainable finance, communication, and sustainability or climate change were identified in the database *Communication & Mass Media Complete*, dating back to 1998. After screening all titles and abstracts as well as using a snowball sampling method, 65 articles were eventually selected that provided insights about message framing and communication strategies on how to engage people with sustainability topics, raise attention, and/or change behavior related to sustainability. While the following suggestions can only be considered a guideline, an overview of general quality standards for advisory talks about ethical funds has already been elaborated on by Schrader [19]. Here, I want to take a step further in discussing the specific communication strategies for SRI that can be employed in each step of the 5A model [25] by financial advisors to make SRI more attractive for private investors and to overcome the challenges and the barriers as outlined above (see Figure 1 for an overview). What should be noted here, however, is that the suggestions for communicative interventions are also dependent on cultural and institutional contexts in which the financial advisors and their clients are embedded. Hence, the following strategies should be considered as general guidelines that can be adjusted according to the given context.

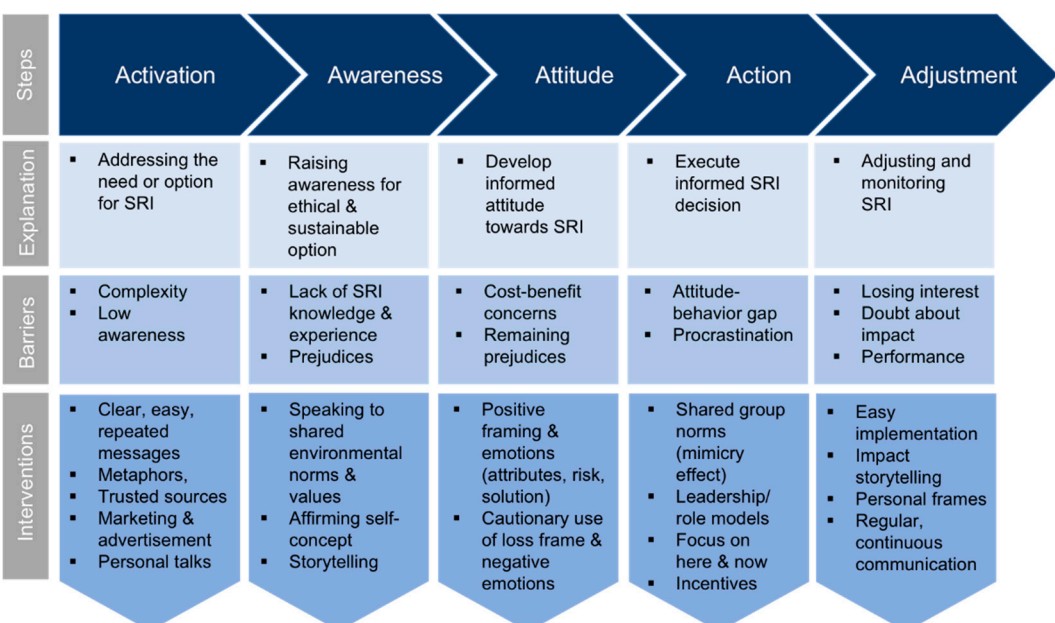

**Figure 1.** The 5A model of the sustainable responsible investing (SRI) decision with strategic communication interventions, adjusted from Pilaj (2017).

### 7.1. Activation

The first step of activating the option for SRI among private investors occurs before the advisory talk usually takes place. Here, the bank or the financial advisor would inquire whether the client is interested in talking about investment opportunities. Another way could be that private investors themselves have started to think about investments and how to manage their money, for example, with regard to their retirement or future plans and, therefore, inquire support from their financial advisors. To overcome the obstacle of complexity at the mental stage of activation [25], the information provided in the first encounters should follow basic principles of successful strategic communication for environmental and sustainable issues, as follows.

Individuals are prone to avoid complex information or information that does not align with their current attitudes (cf. cognitive dissonance, [58]). In addition, previous research has shown that simply providing people with accurate information about climate change and the benefits of pro-environmental behavior will not lead to behavioral change [59,60]. Rather, a single expressive instance (e.g., vivid image or description) is more powerful

than plain statistics for people to draw generalizations [61]. Similarly, the use of imageries and metaphors can make complex concepts more conceivable (cf. [62]), and research has indeed shown that metaphors are successful in conveying persuading messages (e.g., [63]). As Van der Linden and colleagues [64] conclude, "presenting information in a way that is short, simple and easy to comprehend and remember seems to offer the highest probability of success" (p. 261).

With regard to the communication about SRI by financial advisors, this means that, rather than using complex and long descriptions of SRI, advisors should focus on using a clear language and numbers that are easy to understand and that convey compelling information (e.g., risk vs. return). Furthermore, advisors should use descriptions of sustainability projects and investments that are captivating (e.g., metaphors) and supported by appealing visual features. To make this activation more likely, banks and financial advisors could make their clients directly aware of new SRI opportunities in their personal exchanges, for example, via phone, email, or personal talks at the bank. In addition, bank branches and their websites could actively advertise SRI and employ marketing strategies (e.g., on social media, TV, street campaigns) to activate the awareness among private investors for SRI options.

Although clear messages are paramount for comprehension, the human brain learns best by means of repetitions [65]. Underlying this phenomenon is the belief that, the more a message is repeated, the more persuasive this message appears cognitively and the more cognitively available (retrievable) this information becomes [66]. Similarly, it has been argued that more experience and increased frequency of exposure to topics might not only increase understanding but also the likeliness to be open for the ideas presented (e.g., [67,68]). This phenomenon is also closely related to availability heuristics as defined by Tversky and Kahneman [69]. The theory implies that information which has been acquired more recently or more often is easier to retrieve but, in turn, can involve bias probability or frequency estimations.

What is more, a long line of research implies that the credibility of sources of messages is indicative for the intended persuasive effect of messages (e.g., [70]). Hence, establishing trust is of paramount interest when communicating about SRI. Research has shown that, due to opaque information conveyed in green advertisements, consumers attribute low levels of trust to these ads (e.g., [71]). Especially in cases where information is perceived to be confusing, consumers become increasingly skeptical, even leading to the rejection of green products [72]. Hence, following the concept of legitimacy [73], a company gains more credibility for its "green" claims if the firm is indeed involved in sustainable behavior [74] and integrates sustainable norms and values in its organizational behavior (cf. "walking the talk", [75]). Relatedly, research has shown that green/environmental ads should be accompanied by credible sources and informative green/environmental claims in order to be successful (e.g., [76]). In fact, survey research provided evidence that people tend to trust scientists, friends, family, and national research administrations the most when it comes to climate change [77].

Following these findings, financial advisors and the financial institution they are associated with must, at their minimum, convey core values such as sustainability, honesty, and integrity and integrate these values in their organizational behavior to establish trust among their stakeholders (cf. [75,78,79]). To do so, Paetzold and colleagues [20] argue that providers of SRI must present compelling evidence for their credibility and demonstrate that their organization can be trusted (e.g., transparency, official labels and ratings of SRI, membership of SRI alliances, credible auditing of SRI). Furthermore, when presenting claims about the performance or the impact of SRI, financial advisors should support their statements with insights from scientists and official research offices and repeat that information at various occasions [cf. 77]. In short, financial advisors should use "simple clear messages, repeated often, by a variety of trusted sources" ([80], p. 337) when communicating about SRI.

### 7.2. Awareness

In the second step, financial advisors need to make clients aware of SRI options in the advisory talk. Individuals might be very concerned about climate change and the environment (e.g., buying organic food, going by bike) but have not previously thought about whether their money is invested in brown or harmful industries (fossil fuel, coal industry). While the offer of sustainable investment products might strongly depend on the respective financial institute, there are certain communication techniques that the advisors could use to raise awareness among private investors for SRI and to overcome the obstacle of limited attention [25].

The first step could be to make the clients aware of shared values and norms regarding the environment, climate change, and the purpose of finance (e.g., serving society and the environment). A wide array of research in environmental psychology has proven that both personal norms (e.g., [81,82]) and social norms [83,84] are strong drivers to motivate sustainable behavior. While prosocial, altruistic, or biospheric values appear to be strongly related to pro-environmental behavior (e.g., [85,86]), self-interest and beliefs in power and tradition are, on the other side, negatively related with environmentalism [87]. Hence, speaking to those values is crucial, as individuals are driven by the desire to affirm their positive self-concept and seek out information that reinforces their existing beliefs [88].

What is more, based on a review of international research on public opinion on climate change, Wolf and Moser [89] identified storytelling in terms of cultural narratives and social interaction as the most effective method to motivate people to become interested in and stay engaged with climate change. Storytelling—particularly in terms of episodic framing (cf. highlighting the personal aspects of stories, [90])—can thus be effective in raising awareness among investors for SRI. In practice, financial advisors could make use of the prevailing pro-environmental norms among their clients by pointing out that SRI speaks to their concerns about the environment, biodiversity, and climate change and is thus in line with their self-concept. Kidwell, Farmer, and Hardesty [91] found empirical support that fluency (cf. [92]) is the underlying mechanism that leads to spillover effects such as pro-environmental behavior. Hence, if private investors feel their investment behavior is in line with their personal values and norms, it is more likely that they will engage with SRI. Given that personal values are learned, taught, and appropriated through upbringing, education, and social interactions during lifetime [93], financial advisors might simply have to remind customers of their already existing norms and values that are in line with practices of SRI.

### 7.3. Attitude

In the third mental step, a customer's attitude towards SRI should be formed. Thus, financial advisors need to provide convincing and compelling information that will overcome investors' concerns and misconceptions regarding the "lack of information, intransparency, and the perceived lack of impact of SRI" ([25], p. 749). While some of the barriers can be resolved by simply providing information on SRI that is transparent, complete, demonstrates the impact of SRI, and shows scientific evidence that proves that the financial performance of SRI is equally, if not more, successful than conventional investments [41], mere information might not be enough to convince some investors. In fact, previous research implies that information overload in advisory talks can also backfire (e.g., [94]). Hence, rather than simply presenting relevant but stalled information about SRI, financial advisors could make use of framing strategies that present the information in a more convincing and engaging way (cf. storytelling above).

Framing in the field of communication implies highlighting some aspects in messages over others by making them more salient [95]. In the field of environmental studies, three general types of framing have been identified [96,97]. The first type of framing is called attribute framing, in which equivalent information can be framed in either a positive (e.g., the glass is half full) or a negative frame (e.g., the glass is half empty). Risky choice framing is the second type of environmental message framing [97]. It encompasses

the phenomenon that the same level of risk can be framed in either positive or negative terms. The third type in environmental framing research is defined as goal framing [96]. Goal framing deals with the presentation of information that outlines the solutions or the consequences of an issue or an action [95]. Similar to the previous frame categories, the goal frames can either be presented in a positive way (cf. promoting the benefits) or a negative way (cf. highlighting the costs).

Overall, research on whether positive or negative framing is more effective for attitudinal and behavioral change points in opposing directions. Studies have shown, however, that people tend to prefer positive risk wording (e.g., will save 200 people, [98]). Yet, contrary to risk frames, other research has shown that loss frames are generally more effective than gain frames when it comes to framing solutions or consequences (cf. [99]). This behavior can be explained by prospect theory [100], which implies that people are more risk-seeking when facing losses compared to gains. Hence, the loss of $X is considered to be more severe while the equal gain of $X would be considered attractive. Based on these findings, messages about SRI should highlight the low risks and the high gains when investing ethically (e.g., opportunity of saving $X of stranded asset value). However, in some cases, it might also be worth highlighting the risks or the financial losses if climate risks remain ignored in investments (e.g., risk of losing $X if staying invested in fossil fuel).

Another way of framing messages about SRI more effectively in advisory talks is by making use of emotional language. Epstein [101] distinguishes between "two parallel, interacting modes of information processing: a rational system and an emotionally driven experiential system" (p. 709). The two systems continually interact and thereby guide human decision making and judgment [102]. As Epstein [101] summarizes based on a review of numerous studies, knowledge that is derived from the experiential system is often considered more captivating and persuasive in influencing behavior than knowledge derived from the rational system. In addition, psychologists argue that emotions are the direct cause of behavior [103], particularly with regard to sustainable behavior [104]. Hence, speaking to emotions in financial advisory talks could enhance engagement of private investors with SRI. However, which emotions should advisors use in their advisory talks?

In general, it has been argued that positive feelings such as excitement, joy, hope, or enthusiasm induce stronger reactions in terms of acceptance and approach, whereas negative feelings such as sadness or anger rather relate to rejection and flight reactions [105]. Similarly, research on climate change communication recommends avoiding fearful and negative messages [88,106]. Rezvani and colleagues [104] found, for example, that positive affective emotions are more powerful in bringing about pro-environmental behavior than negative affective emotions. However, some environmental research also suggests that negative emotions (e.g., guilt, fear, sadness) drive pro-environmental behavior, although the effect sizes are generally weak (e.g., [107–109]). Thus, facing these difficulties, strategic communication about SRI needs to cautiously weigh the extent to which positive or negative emotions should be revoked. Chapman, Lickel, and Markowitz [110] have argued that emotions should be regarded as an integral part of the cognitive system and that they therefore play a complex role in decision-making processes. Consequently, by making private investors aware of the negative effects of conventional investments on the environment, certain negative emotions (fear, guilt) might motivate them to switch to SRI. Likewise, by speaking to positive emotions such as people's affinity to nature (biophilia) and the positive impact of SRI (joy, enthusiasm) on the environment, investors might equally become more engaged with SRI.

### 7.4. Action

The fourth step is about putting SRI preferences in action. Some private investors might be very interested in SRI and eager to invest but fail to put these intentions in practice. According to Pilaj [25], the most common obstacle in this stage is procrastination, thus postponing decisions into the future. Furthermore, private investors might have the impression that SRI is a complex and time-intensive endeavor and they might therefore

refrain from getting engaged. One way to overcome this barrier is to provide easy, fast, and accessible investment processes. Besides the implementation of easy administration frameworks, financial advisors could make use of additional communication strategies that might bring about the final SRI decision making.

First, humans are prone to copy and mimic behavior of others [111,112]. Given the mimicry instinct of humans [113], people are more likely to engage in environmentally friendly behavior if they believe that many others are doing the same. Thus, the more normal and accepted a certain behavior appears to be in a society, the more likely people will perform the respective behavior [114]. In fact, social influence has been found to be one of the strongest forces to influence sustainable behavior [88,115]. Following Pilaj [25], individuals do not only want to make money, but they also want to "express personal values and seek peer approval" (p. 745). As previous studies have shown, descriptive norms have a considerable impact on sustainable behavior (e.g., [116,117]), particularly when describing group behavior that appears to be spatially closer [118].

Relatedly, humans are inclined to follow the behavior of leaders and people who demonstrate power, wealth, and success [119]. Studies have shown that interventions based on social network influences can positively affect community recycling behavior [120,121]. Hence, by speaking to the mimicry instinct of people, it becomes of paramount interest for financial advisors to highlight increased popularity and success stories of SRI vis-à-vis private investors. What is more, advisors could refer to prominent and powerful leaders in the financial industry who have already committed themselves (more or less) to SRI (e.g., Warren Buffett, Larry Fink, Bill Gates). Lastly, it would be beneficial if advisors showed to what extent SRI has been a successful choice with regards to the respective peer group (e.g., millennials, parents, single households, women).

Second, according to life history theory [122], people tend to discount the future and instead value the present [111,123]. Yet, the topic of sustainability is often perceived as abstract, opaque, and too far away from daily encounters (cf. [124]). It has been argued in psychology, for example, that when temporal distance becomes larger, mental representations become more abstract [125]. White and colleagues [88] recommend matching the communication with a temporal focus, thus making people aware of current dangers of global warming and linking the solution to immediate action. Similarly, research implies that individuals show more engagement in sustainable behavior if they believe that their actions will have a "real" impact (cf. [126]). Following these findings, strategic communication by financial advisors should focus on the impact of SRI that will be palpable in the near future (e.g., renewable energy, building of schools, $CO_2$ capturing). Furthermore, it should be clearly communicated in which projects or organizations peoples' investments will flow and how this can make a difference in the short and the long term (e.g., number of sanitary facilities set up, illnesses prevented, $CO_2$ captured).

Third, behavior is based on habits [127], and accustomed habits are usually difficult to alter [128]. Following Stern's [60] conceptual framework, incentives coupled with promotional information might result in synergistic effects, influencing pro-environmental consumer behavior. As he states, "combining monetary incentives with nonfinancial incentives such as convenience increases program effectiveness" (p. 469). In fact, research has presented evidence that consumers are more likely to engage in prosocial behavior if individuals are rewarded with some form of benefit for themselves [129]. Following this advice, incentives such as monetary rewards or discounts for SRI, or, likewise, a favorable tax treatment (cf. [130]), might work as stimuli for people to engage with SRI.

### 7.5. Adjustment

The last step in bringing about SRI decision is not necessarily a mental step, but it deals with the adjustment and the monitoring of the SRI decision by investors. Hence, after an investor has decided to invest in sustainable and responsible investments, advisors need to assess these investments continuously and keep their clients informed about progress and potential needs for adjustments [25]. In that sense, continuous, transparent, reliable,

and trustworthy information will maintain and strengthen the trust relationships between financial advisors and their clients [79]. However, even here, strategic communication interventions could be helpful to ensure that private investors remain engaged with SRI over a longer period of time. This, for example, includes the provision of regular, transparent, and up-to-date information on the performance and the impact of SRI. Besides the common investor letters and the investment reports, financial advisors and their respective financial institutions could focus more on impact reports that outline in detail how the investments have contributed to environmental, climate, and/or biodiversity fields. This could encompass, for example, a report of how much $CO_2$ emissions have been saved, how much forest and wildlife has been secured, or how many children have profited from food, education, and medicine. By means of visual, personal, and positive stories, advisors could make use of the powerful techniques of framing and storytelling again. Eventually, to help private investors to track their SRI activities and to stay invested and engaged, investment information should be made accessible, easy to implement, and convenient to use (e.g., easy-to-use apps for SRI tracking, intuitive websites). As Maibach [80] contends, to make people put their good intentions in practice, we should "make the behaviors we are promoting easy, fun and popular" (p. 337).

## 8. Summary and Practical Implications

Sustainable responsible investing (SRI) has become a buzz word in the financial sector [2]. Recent numbers imply that financial assets that follow ESG criteria have rapidly picked up in the past years [3] but only amount to about 9% of total European assets. Hence, to fill the investment gap that is needed to finance a low-carbon economy and a sustainable future, scholars and opinion leaders in the financial sector have argued that more capital from private investors is needed to flow into SRI [11–13]. Financial advisors have previously been identified as change agents and multipliers [19,21], hence taking an important role in increasing the market for SRI by making investors aware of sustainable investment opportunities. However, besides remaining misconceptions about SRI and barriers on the side of private investors, financial advisors have attested limited knowledge and activities in the area of SRI. While the demand for training and education at financial institutions regarding SRI is high [24], less is talked about how financial advisors should communicate with private investors to raise more awareness for and engagement with SRI.

Using the 5A model for SRI decision-making [25], the aim of this paper was to show how methods of strategic communication can effectively be used in financial advisory talks to overcome barriers and engage private investors with SRI. Based on a thorough literature review of research related to climate change and sustainability communication, theoretical and practical insights were derived and employed to the context of SRI. For each mental step in the SRI decision-making process (activation, awareness, attitude, action, adjustment), findings from previous research were summarized, evaluated, and put in reference to the communication about SRI in financial advisory talks.

The findings provide a useful perspective and guideline for financial advisors, policymakers, and practitioners in the field of SRI on how to successfully communicate SRI to private investors. Summarizing the practical implications, messages about SRI should first activate the need among private investors to think about investments, and particularly SRI. This should be done by using easy, short, vivid (metaphors), and repeated messages about SRI with trusted sources. In other words, marketing and advertising about SRI should be present at various points of contact with private investors. Furthermore, the financial institutions promoting SRI should make sure to uphold a trustworthy and reliable image regarding sustainability, thus integrating sustainability principles in the very core of their business practices.

In the second step, attention for SRI as an alternative to conventional investments needs to be raised in advisory talks. This can best be achieved by making private investors aware of shared norms and values between SRI and their personal pro-environmental values (e.g., daily sustainable behavior). By reinforcing investors' sustainable self-concept and

using storytelling, advisors might be more successful in raising awareness for SRI. In the third step, the attitude towards SRI needs to be shifted so that concerns and misconceptions about SRI (e.g., high risk and low returns) become overridden. Here, framing strategies such as the use of positive attributes, positive risk, and solution framing can be useful. More specifically, the positive aspects of SRI (returns, impact) but also the potential loss of ignoring SRI (e.g., stranded assets) should be elaborated in advisory talks. The use of positive emotional language, such as hope, joy, and enthusiasm, can be equally effective in positively changing investors' attitude towards SRI, while negative emotions (fear, guilt, sadness) should be used sparsely.

The fourth step is about taking action, meaning that investors eventually decide to engage with SRI. In order to bridge the attitude–behavior gap [11,18], financial advisors should not only remind investors that SRI is in line with common investment behavior in their peer group, they could also point out role models and examples of SRI leadership. What is more, investors might be more inclined to get involved with SRI if they are shown the near-term, palpable impact of SRI. Eventually, easy and simple to implement frameworks for SRI and possible financial incentives (e.g., discount, tax relief) could present the final trigger needed to convince investors about SRI. Eventually, in the last step of adjusting and monitoring SRI, financial advisors should provide their clients with continuous, regular, reliable, and transparent information about their sustainable and responsible investments. Storytelling and impact reports with a personal angle but also sound technologies and online features for tracking investments (e.g., app) should be used to keep investors engaged and enthusiastic about SRI in the long run.

## 9. Limitations and Future Research

One obvious limitation of the strategic communication interventions presented here is that they are based on previous research and theoretical frameworks that have not yet been empirically tested with regard to SRI. What is more, some of these conceptual interventions also need to be adjusted to the respective clients and their investment profile (e.g., risk aversion, time horizon for investments, financial situation, cf. [23]). Given that most of the cited research has been conducted in the Western world, it should also be noted that the communication strategies are open for adjustment depending on the specific social or cultural context. For example, shared environmental norms and values might vary considerably across countries and continents (e.g., nuclear energy as "green" energy in France but not in Germany). Similarly, the use of emotional language in financial advisory talks might not be considered professional or adequate in some countries and could additionally be strongly contingent on the particular advisor–client relationship that has been established and maintained over a long period.

Furthermore, it is important to highlight that, although these communication strategies are meant to open the market for SRI to advance global sustainable development, not all activities in the area of SRI can be trusted in the current market environment [35,36]. Therefore, private investors are equally advised to ask critical questions about SRI and to check whether the selected investments are in line with their own environmental values and norms. Similarly, this means that financial institutions need to be transparent and open about their investments, selection criteria for sustainable and responsible investments, and their evaluation of performance and impact of SRI. Eventually, it should be free to decide by private investors themselves whether SRI presents the adequate way to them to invest in a more sustainable future. Alternatives could be sustainable saving plans, impact investment, charity or donation, or simply banking with ethical and green banks.

However, given that SRI is still a better investment option compared to conventional investments (e.g., fossil fuel, tobacco, ammunition), the potentials of SRI should be seized based on the communication interventions provided here. To gain more insights into the effectiveness of the described communication strategies, future research should study the various interventions and their effects on attitudinal and behavioral change among private investors regarding SRI. While most of the framing strategies would best be tested in an

experimental setting or by survey experiments, observational studies of financial advisory talks coupled with interviews and surveys could also provide in-depth insights into the effectiveness of certain communication strategies in the SRI decision making process. Until this empirical research has been conducted and tested in the future, the insights gained from this literature review and theoretical discussion will offer useful, hands-on guidelines for practitioners and academics alike who seek to better understand how to convey SRI in the financial and public sector.

**Funding:** This research was funded by the European Union's Horizon 2020 research and innovation programme under the Marie Skłodowska-Curie grant agreement no. 834638.

**Institutional Review Board Statement:** Not applicable.

**Informed Consent Statement:** Not applicable.

**Data Availability Statement:** Not applicable.

**Conflicts of Interest:** The author declares no conflict of interest.

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
