# Peer review of "Communicating Sustainable Responsible Investments as Financial Advisors: Engaging Private Investors with Strategic Communication"

_sustainability, doi:10.3390/su13063161_

Round 1
Reviewer 1 Report
The paper offers a high-quality insight into how to communicate SRI by financial advisors and in general by the financial sector. It is well written and adequately settled within the literature. As a such, it is worth to be published in its current form.
However, there are some small points for discussion or some improvements in the article. The first is, although there is a concluding remark that the proposed model may depend on cultural and institutional context, it would be advisable to introduce such issues much earlier into the discussion. There is a question on the applicability of the model in various geographical/cultural contexts - would be it different in Nairobi or Tokyo than in London? There is also the issue of SRI as an umbrella term for so different types of investments by their size, subject or geography - are these factors influencing the proposed model or communication strategy?
PS. There are some small misspellings (e.g. line 609 there is "ction" (action).
Author Response
RESPONSE TO REVIEWER 1 COMMENTS
COMMENT: The paper offers a high-quality insight into how to communicate SRI by financial advisors and in general by the financial sector. It is well written and adequately settled within the literature. As a such, it is worth to be published in its current form.
RESPONSE: Dear Reviewer 1, thank you very much for your positive evaluation of my manuscript. I am delighted to hear that you consider it “worth to be published in its current form.”
COMMENT: However, there are some small points for discussion or some improvements in the article. The first is, although there is a concluding remark that the proposed model may depend on cultural and institutional context, it would be advisable to introduce such issues much earlier into the discussion.
RESPONSE: I agree with your comment to introduce the point of cultural and institutional contingencies of the model earlier into the discussion. I have introduced the issue in the beginning of the discussion of strategic interventions (see p. 7).
COMMENT: There is a question on the applicability of the model in various geographical/cultural contexts - would be it different in Nairobi or Tokyo than in London?
RESPONSE: Similar to the point mentioned above, I also agree that the geographical and cultural dependency of the model should have been made clearer in the manuscript. I have added information on the cultural and geographical contingencies of the model in the “Limitation and Future Research” section (see p. 13).
COMMENT: There is also the issue of SRI as an umbrella term for so different types of investments by their size, subject or geography - are these factors influencing the proposed model or communication strategy?
RESPONSE: Thank you for this comment. Indeed, I should have pointed out that the term of SRI is controversial and that similar, related terms are used across the market. Although the concepts, terms and definitions might vary slightly, it does not necessarily influence the proposed model for strategic communication interventions by financial advisors. In fact, it makes effective and clear communication about SRI even more necessary. I have added this information in the literature review (see pp. 2-3).
COMMENT: PS. There are some small misspellings (e.g. line 609 there is "ction" (action).
RESPONSE: Thank you for this note. I have corrected the spelling mistakes.

Reviewer 2 Report
There is no research. This is a description or extension of another author's study. That is not enough for a scientific article.
Author Response
RESPONSE TO REVIEWER 2 COMMENTS
COMMENT: There is no research. This is a description or extension of another author's study. That is not enough for a scientific article.
RESPONSE: While I appreciate the critical stance taken by Reviewer 2, I would like to point out that an explicit aim of the journal “Sustainability”, as stated on the website (https://www.mdpi.com/journal/sustainability/about), is to “encourage scientists to publish their experimental and theoretical research (highlighted by me) relating to natural sciences, social sciences and humanities in as much detail as possible in order to promote scientific predictions and impact assessments of global change and development.” Furthermore, the “About” section of the journal states that there are “unique features” of Sustainability, for example: “manuscripts regarding research proposals and research ideas will be particularly welcomed.” Hence, based on my theoretical discussion and presentation of a communication model for financial advisory talks about SRI with an outlook for future research, I do believe that my manuscript fits the scope of Sustainability.
What is more, the description of the section “Economic and Business Aspects of Sustainability” says explicitly that this section addresses “theoretical analyses explaining new phenomena or puzzles, or the development of theoretical or empirical methods likely to be useful for further research–all within the effects of sustainability issues.” In addition, one of the main scopes of this section is to “stimulate cross-sectoral exchanges for the development of…new awareness towards a new sustainability-oriented society.” Eventually, the section also states that it is interested in conceptual articles.
While I do agree that I did not collect new empirical data for this manuscript, I do believe that a literature review with a critical discussion and interdisciplinary extension of a conceptual model so that it can be employed both in research and practice, is of high scientific and social value; and thus also of value for Sustainability. To conclude, I strongly believe that my manuscript is enough to be considered a scientific article.
Reviewer 3 Report
General comment.
There is an interesting idea with this article. While there is a need to boost SRI to meet the SDGs, it is well acknowledged that actual achievements are dramatically lagging, and more should be done to attract private and institutional funds.
However, and in order to get a better impact, my recommendation is to rewrite the first section on SRI, to better reflect the debates and issues related to it, and also to better structure the part on communication, leaving aside elements that are more at a level of a user’s guide.
I elaborate on this below.
More on the article
This article is presenting a review of the SRI scene, on one side, and of communication techniques, or rather persuasive techniques on the other side, with the aim of proposing a methodology, or a tool for achieving the goal of mobilizing the necessary SRI’s.
However, I think that the article would gain more impact by revisiting both sections:
- On one side, banks and finance typically classify SRI under 1) exclusion criteria (no coal, tobacco, etc…), 2) ESG rating (how well does a firm behave or comply with this), and 3) impact investing. This matters, because the acceptance will differ accordingly.
- On exclusion criteria, this is simple and straightforward, and this makes a large majority of so-called SRI. But on the other side, it tells little about the attitude and engagement of the company, once has not been excluded.
- Selecting a company on their ESG rating is supposed to better reflect how well the company is performing in terms of ESG. But the problem here is that we do have issues with the rating. As a beginning, you can look at an article of the FT: ‘’Lies, damned lies and ESG rating methodologies’’.
- And then, we have impact investing. Here the problem has to do with the impact assessment, a whole topic in itself, as it is torn between lack of reproducibility and the sheer complexity of the process. Here, financial institutions typically use the concepts of tier 1, 2, 3 according to the level of circumscription of the analysis.
This is important because investors are often reluctant to embark on SRI, as they see clearly the loopholes of the approach, and are therefore reluctant to advise clients accordingly. A lot of initiatives are presently underway, such as the discussion on integrated reporting, a substantial part of the maturing of SRI.
An additional element of the discussion is about the profitability of SRI’s. The jury is still a bit out on this, but to a large extent, the view will also depend on the type of SRI:
- exclusion criteria, on average, the stocks for fossil fuel-based companies or tobacco companies did not perform well in 2020 or in the last 3 years. You can do some search on this as a confirmation.
- ESG, when criteria are wobbling (see above), drawing conclusions may be daring, but one point is clear: companies that do have a long term strategy, and as such embed ESG criteria, tend to do better, probably as an indicator of better management.
- impact, there is little research, and therefore evidence, as the methodology is not universal, and therefore such companies are rare.
But I really think that your article should elaborate on these elements in order to gain credibility and impact, as these are hot topics of discussion among the Sustainable Finance community.
Otherwise, the text on setting the scene of SRI is OK, albeit it contains a lot of ‘reinforcing statements’ that turn ultimately into redundancies.
With respect to financial advisors, keep in mind how banks are functioning: the financial advisor is receiving typically on a weekly basis a list of products to place to his clients, and his incentives are based on such placements and the profitability of his portfolio of customers!
- The section on ‘strategic Communication….’ is a review of tools and techniques of persuasion that a financial advisor could use in order to place more SRI. It is based on the 5A model of Pilaj. The article is describing this 5a methodology and the nudging tools. But before that, the reader would be interested to know why this methodology was selected, and what benefits it does provide. Keep in mind that this model is not widely used outside academic circles. Probably, an overview of similar techniques, starting from the behavioral science and the persuasive technologies, would help. Otherwise, some classical salesmen techniques would apply here as well, and financial advisors (at least in large banks), are trained on this either. Therefore, it would help to clarify what the 5A model would bring.
On the content of the section, a lot has to do with communication recipes, typically encountered in training for e.g. sales workforces. It contains also many repetitions, and to gain more impact, it should be ideally restructured. E.g., lines 551-ff on storytelling: this is a common concept in communication, but should it be under ‘adjustment’ or rather ‘Awareness’?
Author Response
RESPONSE TO REVIEWER 3 COMMENTS
COMMENT: There is an interesting idea with this article. While there is a need to boost SRI to meet the SDGs, it is well acknowledged that actual achievements are dramatically lagging, and more should be done to attract private and institutional funds.
RESPONSE: Thank you very much for your comment. I agree that there is a strong need in attracting private and institutional capital to boost SRI. I hope this paper can offer a guideline for financial advisors to seize these potentials.
COMMENT: However, and in order to get a better impact, my recommendation is to rewrite the first section on SRI, to better reflect the debates and issues related to it, and also to better structure the part on communication, leaving aside elements that are more at a level of a user’s guide.
I elaborate on this below.
RESPONSE: I agree that recent debates and issues related to SRI should have been included in the literature review and that some parts of the communication section were repetitive. I have responded to your detailed comments below. Thank you for your suggestions.
COMMENT: This article is presenting a review of the SRI scene, on one side, and of communication techniques, or rather persuasive techniques on the other side, with the aim of proposing a methodology, or a tool for achieving the goal of mobilizing the necessary SRI’s.
However, I think that the article would gain more impact by revisiting both sections:
On one side, banks and finance typically classify SRI under 1) exclusion criteria (no coal, tobacco, etc…), 2) ESG rating (how well does a firm behave or comply with this), and 3) impact investing. This matters, because the acceptance will differ accordingly. On exclusion criteria, this is simple and straightforward, and this makes a large majority of so-called SRI. But on the other side, it tells little about the attitude and engagement of the company, once has not been excluded. Selecting a company on their ESG rating is supposed to better reflect how well the company is performing in terms of ESG. But the problem here is that we do have issues with the rating. As a beginning, you can look at an article of the FT: ‘’Lies, damned lies and ESG rating methodologies’’. And then, we have impact investing. Here the problem has to do with the impact assessment, a whole topic in itself, as it is torn between lack of reproducibility and the sheer complexity of the process. Here, financial institutions typically use the concepts of tier 1, 2, 3 according to the level of circumscription of the analysis. This is important because investors are often reluctant to embark on SRI, as they see clearly the loopholes of the approach, and are therefore reluctant to advise clients accordingly. A lot of initiatives are presently underway, such as the discussion on integrated reporting, a substantial part of the maturing of SRI.
RESPONSE: I absolutely agree with Reviewer 3 that much more could have been said about the classification of SRI and therewith the respective issues that arise. I also thank the Reviewer for pointing me to the FT article, which was very interesting and certainly of use for this article. I have followed the Reviewer’s advice and adjusted the section about “Challenges of SRI,” including the classifications of SRI and some of the issues that the industry is currently facing. I have also added more information on the current developments in the area of integrated reporting (see p. 4).
COMMENT: An additional element of the discussion is about the profitability of SRI’s. The jury is still a bit out on this, but to a large extent, the view will also depend on the type of SRI: exclusion criteria, on average, the stocks for fossil fuel-based companies or tobacco companies did not perform well in 2020 or in the last 3 years. You can do some search on this as a confirmation. ESG, when criteria are wobbling (see above), drawing conclusions may be daring, but one point is clear: companies that do have a long term strategy, and as such embed ESG criteria, tend to do better, probably as an indicator of better management. impact, there is little research, and therefore evidence, as the methodology is not universal, and therefore such companies are rare. But I really think that your article should elaborate on these elements in order to gain credibility and impact, as these are hot topics of discussion among the Sustainable Finance community. Otherwise, the text on setting the scene of SRI is OK, albeit it contains a lot of ‘reinforcing statements’ that turn ultimately into redundancies.
RESPONSE: Thank you for these insights on the profitability of SRI. I have added some more recent findings on the profitability of SRI and the distinctions to be made. I have also tried to eliminate redundancies and “reinforcing statements” in the literature review.
COMMENT: With respect to financial advisors, keep in mind how banks are functioning: the financial advisor is receiving typically on a weekly basis a list of products to place to his clients, and his incentives are based on such placements and the profitability of his portfolio of customers!
RESPONSE: That is a good point, thank you. I have taken this up in the literature review regarding the challenges of financial advisors.
COMMENT: The section on ‘strategic Communication….’ is a review of tools and techniques of persuasion that a financial advisor could use in order to place more SRI. It is based on the 5A model of Pilaj. The article is describing this 5a methodology and the nudging tools. But before that, the reader would be interested to know why this methodology was selected, and what benefits it does provide. Keep in mind that this model is not widely used outside academic circles. Probably, an overview of similar techniques, starting from the behavioral science and the persuasive technologies, would help. Otherwise, some classical salesmen techniques would apply here as well, and financial advisors (at least in large banks), are trained on this either. Therefore, it would help to clarify what the 5A model would bring.
RESPONSE: I very much appreciated this comment. Indeed, I should have offered more insights into why the 5A model was chosen over others and what benefits it provides for the purpose of my argumentation. I have included a separate chapter “Theoretical Models for Moral and Ethical Investment Behavior” with a short overview of common models and theories used in environmental research as well as an overview of models that have been used in the field of SRI. I also make clear why these models fall short with regard to the purpose of this paper and why the 5A model by Pilaj fits much better as a framework (see p. 6).
COMMENT: On the content of the section, a lot has to do with communication recipes, typically encountered in training for e.g. sales workforces. It contains also many repetitions, and to gain more impact, it should be ideally restructured. E.g., lines 551-ff on storytelling: this is a common concept in communication, but should it be under ‘adjustment’ or rather ‘Awareness’?
RESPONSE: Thank you for this suggestion. I have reviewed this section and tried to eliminate repetitions and redundancies. I also reshuffled the part of “storytelling” and put it in the section dealing with “Awareness” and only mention it in passing in the part on “Adjustment”. Thank you for this recommendation.
Round 2
Reviewer 3 Report
There are substantial additions to the original text, and the document has gained value.
I would still suggest making clear that the article is not really a research stricto-senso, and should be rather viewed as a perspective on communicating SRI, and readers should not expect an experiment on a focus group, with statistical processing and (in-)validation of hypothesis.
As a 'perspective' this document can offer an interesting view.
Author Response
Response Reviewer 3
COMMENT: There are substantial additions to the original text, and the document has gained value.
RESPONSE: Thank you very much for your positive feedback. I appreciate that very much.
COMMENT: I would still suggest making clear that the article is not really a research stricto-senso, and should be rather viewed as a perspective on communicating SRI, and readers should not expect an experiment on a focus group, with statistical processing and (in-)validation of hypothesis. As a 'perspective' this document can offer an interesting view.
RESPONSE: I agree with you. I have introduced the fact that this is rather a “perspective” than an empirical study. I have made adjustments to that in the abstract, introduction and discussion/conclusion, pointing out that this manuscript is a theoretical and conceptual discussion.